# Elucidating the immune infiltration in acne and its comparison with rosacea by integrated bioinformatics analysis

**Lu Yang⍟, Yan-Hong Shou⍟, Yong-Sheng Yang\*, Jin-Hua Xu\***

Department of Dermatology, Huashan Hospital, Fudan University, Shanghai, China

⍟ These authors contributed equally to this work.
\* yangyongsheng73512@126.com (YSY); jinhuaxu@fudan.edu.cn (JHX)

**Data Availability Statement:** All relevant data are within the Supporting Information files. A list of the relevant data accession numbers can be found within the Supporting Information file.

## Abstract

### Background

Acne vulgaris and rosacea are common inflammatory complications of the skin, both characterized by abnormal infiltration of immune cells. The two diseases can be differentiated based on characteristic profile of the immune cell infiltrates at the periphery of disease lesions. In addition, dysregulated infiltration of immune cells not only occur in the acne lesions but also in non-lesional areas of patients with the disease, thus characterizing the immune infiltration in these sites can further enhance our understanding on the pathogenesis of acne.

### Methods

Five microarray data-sets (GSE108110, GSE53795, GSE65914, GSE14905 and GSE78097) were downloaded from Gene Expression Omnibus. After removing the batch effects and normalizing the data, we applied the CIBERSORT algorithm combined with signature matrix LM22, to describe 22 types of immune cells' infiltration in acne less than 48 hour (H) old, in comparison with non-lesional skin of acne patients, healthy skin and rosacea (including erythematotelangiectatic rosacea, papulopustular rosacea and phymatous rosacea) and we compared gene expression of Th1 and Th17-related molecules in acne, rosacea and healthy control.

### Results

Compared with the non-lesional skin of acne patients, healthy individuals and rosacea patients, there is a significant increase in infiltration of neutrophils, monocytes and activated mast cells around the acne lesions, less than 48 H after their development. Contrarily, few naive CD4+ T cells, plasma cells, memory B cells and resting mast cells infiltrate acne sites compared to the aforementioned groups of individuals. Moreover, the infiltration of Regulatory T cells (Tregs) in acne lesions is substantially lower, relative to non-lesional sites of acne patients and skin of healthy individuals. In addition, non-lesional sites of acne patients exhibit lower infiltration of activated memory CD4+ T cells, plasma cells, memory B cells,

**Funding:** The authors received no specific funding for this work.

**Competing interests:** The authors have declared that no competing interests exist.

M0 macrophages, neutrophils, resting mast cells but higher infiltration of Tregs and resting dendritic cells relative to skin of healthy individuals. Intriguingly, we found that among the 3 rosacea subtypes, the immune infiltration profile of papulopustular rosacea is the closest to that of acne lesions. In addition, through gene expression analysis of acne, rosacea and skin tissues of healthy individuals, we found a higher infiltration of Th1 and Th17 cells in acne lesions, relative to non-lesional skin areas of acne patients.

## Conclusions

Our study provides new insights into the inflammatory pathogenesis of acne, and the difference between acne and rosacea, which helps in differentiating the two diseases. Our findings also guide on appropriate target therapy of the immune cell infiltrates in the two disease conditions.

## 1. Introduction

Acne vulgaris (often called acne), which affects the pilosebaceous unit, is one of the most common inflammatory disorder of skin mainly in adolescents and young adults [1]. The skin lesions are predominant on the face but can also appear on the neck, back and chest. On the other hand, rosacea is a common chronic inflammatory cutaneous disorder which often occurs in the central facial skin of individuals between 30–50 years old. In addition, rosacea is more common in women than men. Clinically, acne is characterized by the formation of comedones, papules, nodules, pustules and cysts on the skin, whereas rosacea presents with recurrent flushing facial erythema, pustules, papules and telangiectasia [2]. The two complications adversely affect the patient's appearance. This may embarrass the affected individual, lower their self-esteem and the overall quality of life, and in worst cases, lead to depression. In adolescents, the prevalence of acne ranges from about 40% to 90% [3, 4], and less than 1% to 22% in the general population [5].

The precise pathogenesis of acne and rosacea remains unclear. However, research shows that both diseases are associated with microbial infection-associated inflammation. For instance, acne is often triggered by *Corynebacterium acnes* (*C. acnes*), *Staphylococcus epidermis* and *Malassezia furfur* infections in the pilosebaceous units, whereas rosacea is induced by *Demodex folliculorum*, *Staphylococcus epidermidis*, *Helicobacter pylori* and *Bacillus oleronius* infections in skin. Over-secretion of sebum resulting from hyperproliferation of follicular epidermis and the circulation of dehydroepiandrosterone (DHEA) has been implicated in the development of acne. Besides inflammation, other factors such as genetic make-up, smoking, eating habits (overeating junk food, dairy products) and usage of oil-based cosmetics [3, 6] increases the risk of developing acne. Ultraviolet (UV) light radiation, neurogenic dysregulation, decreased barrier function of skin and dysregulated vascular function also participate in the pathogenesis of rosacea [7].

Dysregulation of the innate and adaptive immune systems are thought to be central in the pathogenesis of acne and rosacea. For instance, abnormal infiltration of polarized Th1/Th17 cells, macrophages and mast cells has been observed in sites in the periphery of acne and the main subtypes of rosacea lesions (such as erythematotelangiectatic rosacea, papulopustular rosacea and phymatous rosacea). In addition, infiltration of neutrophils has been observed in early developmental stages of acne as well as papulopustular and phymatous rosacea. On the

other hand, B and plasma cells are involved in the late stage of acne as well as papulopustular and phymatous rosacea development [8, 9]. Even though the pattern of upregulated gene expression is comparable between acne and rosacea lesions [9, 10], the precise profile of inflammatory cell infiltration between the two diseases remains to be elucidated. In addition, it is not clear whether the dysregulated immune infiltration occurs in skin distant to acne lesions. Thus defining the immune cells infiltration profile can not only unravel the immunopathogenesis of acne and rosacea, but can further reveal novel therapeutic targets for the diseases.

CIBERSORT is a computational tool widely used in the analysis of immune responses of 22 types of immune cells under numerous diseases such as atopic dermatitis [11], lupus nephritis [12], ischemia-reperfusion injury in renal transplantation [13], osteoarthritis [14, 15], in fresh, frozen and fixed specimens [16] based on microarray and RNA-seq data. However, only few researches have utilized the CIBERSORT method to investigate immune cell infiltration characteristics in acne and rosacea.

In the present study, we analyzed gene expression profiles of lesions (onset within 48 H) and normal skin of these acne patients, lesions of 38 rosacea patients and normal skin of 27 healthy individuals. We then analyzed the profile and differences in immune cells between acne and their corresponding non-lesional skin, rosacea as well healthy individuals. Findings of this research may broaden our understanding of inflammatory events leading to acne and possible therapeutic options.

## 2. Materials and methods

### 2.1. Microarray data

Gene expression profiles of GSE108110, GSE53795, GSE65914, GSE14905 and GSE78097 were downloaded from GEO database (https://www.ncbi.nlm.nih.gov/geo/) (GPL570 platform, Affymetrix Human Genome U133 Plus 2.0 Array), we chose 36 samples in GSE108110 including 18 acne patients' biopsies in papule less than 48 H old and 18 biopsies in their non-lesional skin. GSE53795 contains 24 samples, 12 biopsies in acne patients' papule less than 48 H old and 12 biopsies in their non-lesional skin. In GSE65914, 38 rosacea patients' biopsies, including 14 erythematotelangiectatic rosacea, 12 papulopustular rosacea and 12 phymatous rosacea were chosen, 21 biopsies from healthy people in GSE14905 and 6 in GSE78097 were also included. We used Bioconductor packages (http://www.bioconductor.org/) and R software (version 3.6.3; https://www.r-project.org/)) in the data analyses. Raw data of the gene expression profiles were read through the "*affy*" package [17], the inter-batch differences were removed using the "*sva*" package [18]. We visualized the effect of removing inter-batch difference in the quantile–quantile plot (Q–Q plot), and a two-dimensional PCA cluster plot was drawn to show the effect of inter-sample correction.

### 2.2. Evaluation of immune cell infiltration by CIBERSORT analysis

The normalized gene expression data was uploaded to CIBERSORT. We chose the samples with $P < 0.05$ from CIBERSORT algorithm, and obtained the 22 kinds of immune cells' composition. These immune cells included naive CD4+ T cells, activated memory CD4+ T cells, resting memory CD4+ T cells, CD8+ T cells, regulatory T cells (Tregs), follicular helper T cells, gamma delta T cells, naive B cells, plasma cells, memory B cells, M0 macrophages, M1 macrophages, M2 macrophages, activated dendritic cells, resting dendritic cells, activated mast cells, resting mast cells, activated NK cells, resting NK cells, monocytes, neutrophils and eosinophils. CIBERSORT was run in the mode of absolute quantification (method: sig score), rather than the relative mode used in the original publication. We used "absolute CIBERSORT score" to reflect the absolute content of immune cells in each sample. The proportion of 22 kinds of

immune cells in the acne and normal skin tissue was reflected by the barplot when the total infiltration of immune cells was regarded as 100%. Then, based on absolute CIBERSORT score of 22 kinds of immune cells in each sample, we used "*ggplot2*" package to draw violin diagrams ($P < 0.05$ was considered statistically significant) and two-dimensional PCA maps to identify if there were differences in the immune cell infiltration among groups. We used the "OriginPro (2021)" software to visualize the correlation between immune cells; "*pheatmap*" package [19, 20] was utilized to draw heatmaps to show the different immune cell infiltration.

### 2.3. The expressions of activated mast cell related molecules

We evaluated gene expression of activated mast cell related molecules including TAC1, TACR1, ADCYAP1, ADCYAP1R1, VIP, VIPR1, VIPR2, HRH1, HRH2, HRH3, HRH4, TPSAB1 and TPSB2. Non-parametric Kruskal-Wallis test for comparison was used among the four groups, multiple comparisons were corrected with the Bonferroni method followed by non-parametric Wilcoxon rank-sum test for pair-wise comparisons among groups. These analyzes were performed by STATA 13.0 (College Station, Texas 77845 USA), adjust $P < 0.004$ was considered statistically significant and all the data was visualized by GraphPad Prism 8.4.2.

### 2.4. Th1 and Th17 cells infiltrating analysis

The analysis of 22 kinds of immune cells through CIBERSORT algorithm did not include Th1 and Th17 cells, thus we evaluated gene expression of Th1-related molecules such as IL2, IL12RB1 and Th17-related cytokines, including IL17A, IL17F in acne less than 48 H old, non-lesional skin of acne patients, rosacea and healthy individuals. The same statistical and data visualization methods were used as 2.3.

## 3. Results

### 3.1. Source and organization of the data

Relevant data extracted from GSE108110, GSE53795, GSE65914, GSE14905 and GSE78097 datasets was merged together after removing the inter-batch differences among the repositories as shown in the Q-Q plot (Fig 1). The two-dimensional PCA cluster plot revealed that the degree of immune cells infiltration in healthy individuals was between that of lesioned and non-lesioned skin of acne patients. In addition, there was a clear distinction in the infiltration of immune cells among the three rosacea subtypes. Moreover, the immune profile for patients with acne for less than 48 H was comparable to that of patients with papulopustular rosacea (Fig 2A and 2B).

### 3.2. The patterns of immune cell infiltration

CIBERSORT algorithm was applied to investigate the different immune cell infiltration among groups. We got absolute CIBERSORT scores which could reflect the absolute contents of 22 kinds of immune cells in each sample. According to the absolute CIBERSORT scores, we calculated when the total infiltration was 100%, the proportions of each kind of infiltrated cells. Fig 3A showed the patterns of 22 subpopulations of immune cells' proportion in acne less than 48 H and healthy control. Fig 3B illustrates the proportional infiltration patterns of the 22 immune cell populations in non-lesional skin of acne patients and healthy individuals. Overall, CD4+ T cells, dendritic cells and macrophages were the highest infiltrating cells in skin. Meanwhile, the proportions of infiltrated neutrophils, activated mast cells and M0 macrophages were highest in acne less than 48 H group, the healthy individuals came second, whereas the distant areas from acne lesions displayed the lowest infiltration of the above immune cells. The

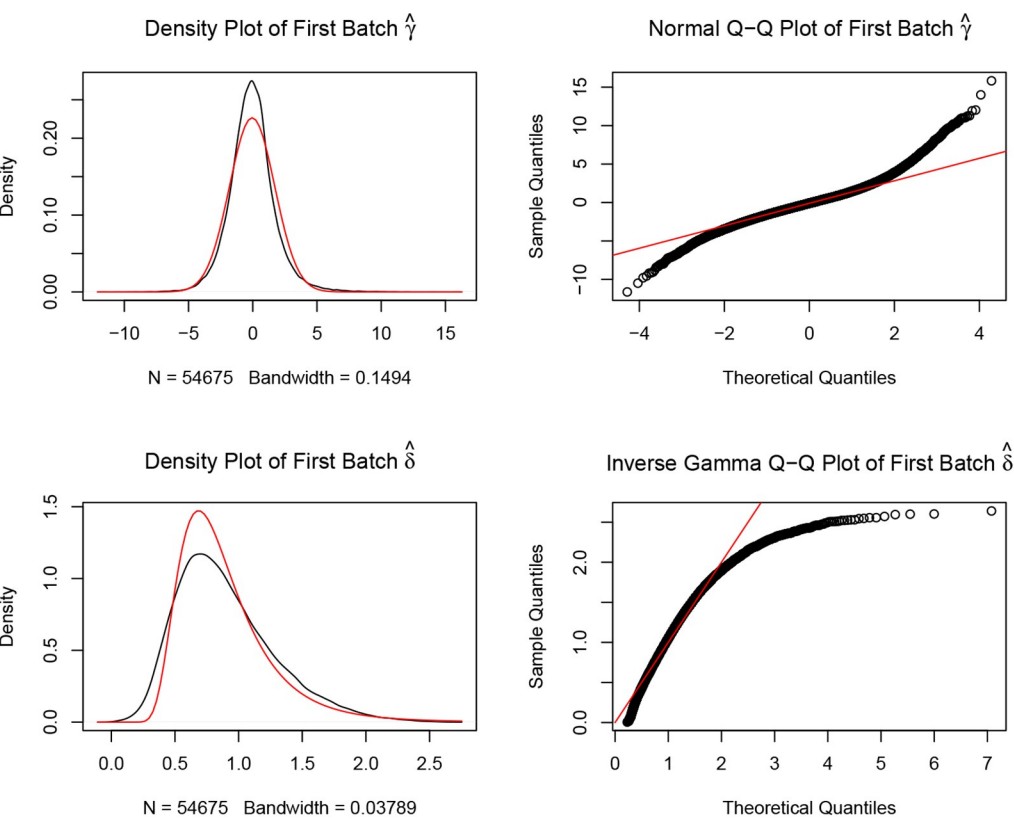

**Fig 1. Removing the inter-batch differences.** The actual density map of data distribution was expressed by the black line, the theoretical density map of data distribution was expressed by the red line. The black circles represent quantiles corresponding to the same cumulative probability.

heat maps for infiltration of immune cells between acne patients and healthy individuals as well as between non-lesional skin of acne and healthy patients is shown in Fig 3C and 3D, respectively.

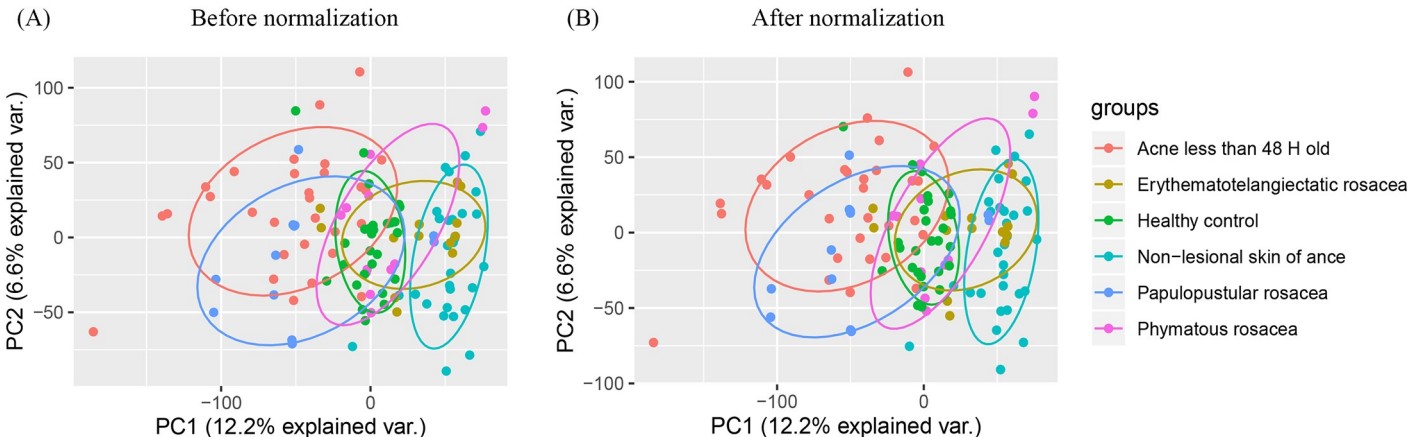

**Fig 2. Normalization of the datasets.** (A) A two-dimensional PCA cluster plot before normalization. Although there is a substantial distance between samples from the same group, there is a clear relationship among the groups. (B) PCA cluster plot after normalization. Samples from the same group were more concentrated in the same region. There was also a clear intergroup relationship.

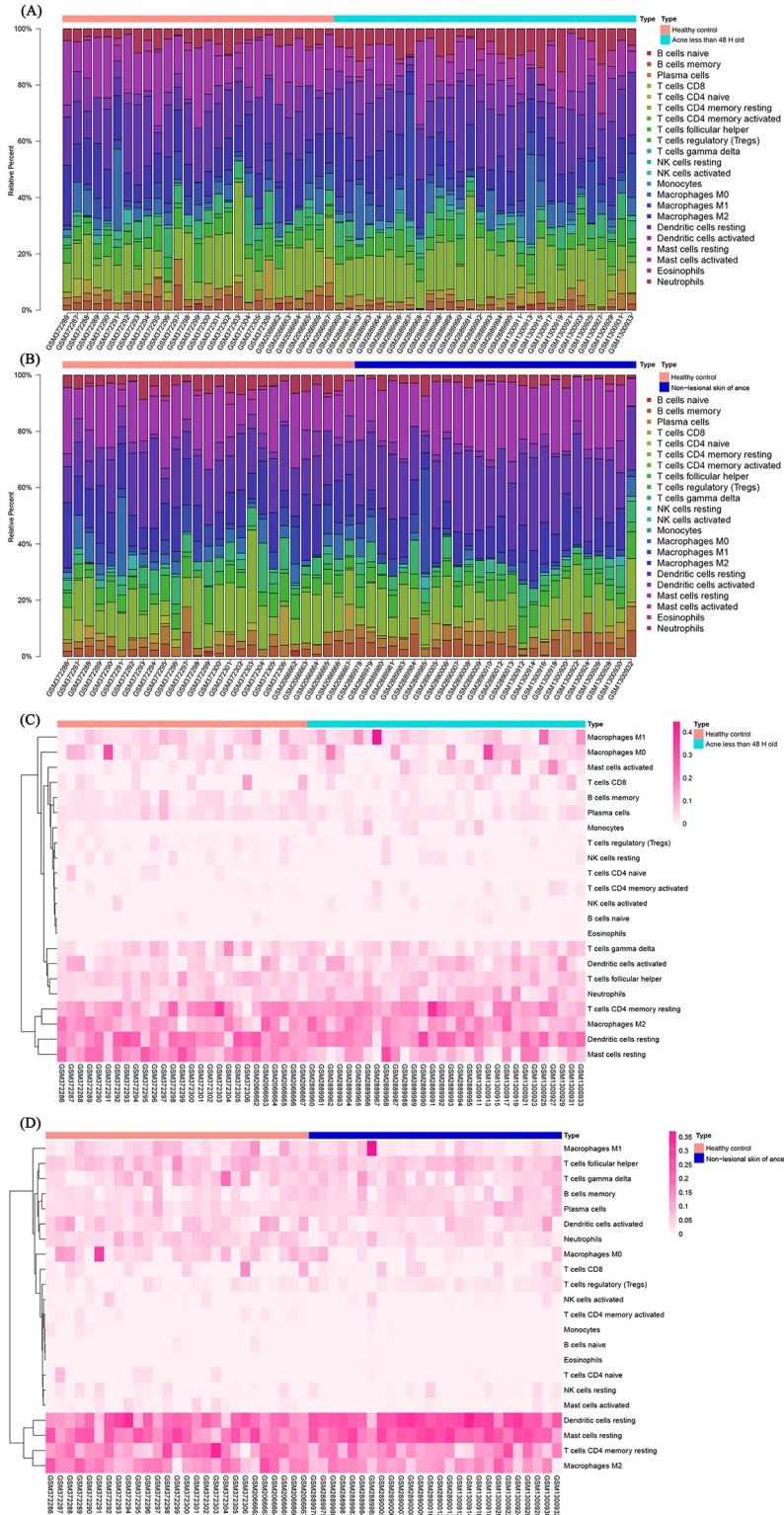

**Fig 3. The infiltration pattern of immune cells among different groups.** The infiltration patterns of 22 subpopulations of immune cells in (A) acne less than 48 H and healthy individuals and (B) non-lesional skin of acne patients and healthy individuals. Heat map for immune cells infiltration between (C) acne lesions and healthy individuals and (D) non-lesional skin tissues of acne patients and healthy individuals. Blue line represents the acne group, dark blue line represents non-lesional skin of acne patients, whereas the red line represents the healthy group.

### 3.3. The correlation between infiltrated immune cells

As is shown in Fig 4, by analyzing the absolute CIBERSORT scores of acne less than 48 H group, there was a positive strong correlation between activated memory CD4+ T cells and M1 macrophages, neutrophils and activated mast cells, activated memory CD4+ T cells and activated mast cells as well as plasma cells and memory B cells in the acne group. On the other hand, there was a negative correlation between the infiltration of resting memory CD4+ T cells and follicular helper T cells, resting mast cells and both activated mast cells neutrophils (*P*<0.001).

### 3.4. The differences in immune cell infiltration between groups

Violin plot of absolute CIBERSORT scores revealed that compared to non-lesional acne patients, healthy individuals and rosacea patients, the acne lesions exhibited higher infiltration of neutrophils, monocytes and activated mast cells, but lower infiltration of naive CD4+ T cells, plasma cells, memory B cells and resting mast cells. Meanwhile, there was a lower infiltration of Tregs in the acne lesions than non-lesional sites of acne patients and skin of healthy individuals. Compared to healthy individuals, there was a lower infiltration of activated memory CD4+ T cells, plasma cells, memory B cells, M0 macrophages, neutrophils and resting mast cells but a higher infiltration of Tregs and resting dendritic cells in non-lesional skins of acne patients (Fig 5A–5D).

### 3.5. Molecules associated with infiltration of activated mast cells

Activated mast cells were one of the populations over-recruited in acne lesions relative to skin of healthy individuals or other skin conditions (*P*< 0.004). Intriguingly, the recruitment of mast cells was accompanied by secretion of numerous molecules such as substance P (SP), a neuropeptide associated with stress-induced acne, which stimulates mast cells degranulation and macrophage infiltration [21]. Even though SP has insignificant roles in rosacea, other neuropeptides such as pituitary adenylate cyclase-activating polypeptide (PACAP) or vasoactive intestinal peptide (VIP) which stimulates mast cells to release histamine, tryptase and other inflammatory mediators as well participate in the induction of the itchy and burning sensations [22] were over-expressed in rosacea lesions. Further analysis revealed that TAC1 and TACR1 encode SP and its receptor, respectively, whereas ADCYAP1 and ADCYAP1R1 encode PACAP and its receptor, respectively. VIP and its receptor are encoded by VIP and VIPR1, VIPR2, whereas histamine is encoded by HRH1-4. TPSAB1 and TPSB2 encode tryptase. As shown in Fig 6, there was an upregulated expression of SP and PACAP in acne lesions relative to normal skins from the same acne patients. VIP and histamine were expressed highest in acne lesions, moderate in rosacea lesions and skin of healthy patients and least in normal skins of the acne patients. There were no significant differences in the expressions of tryptase genes among the four groups.

### 3.6. The infiltration of immune cells between acne lesions and lesions of the three rosacea subtypes

PCA cluster analysis results which were based off the immune cell infiltration data revealed that the pattern of immune infiltration in acne lesions was comparable to that of papulopustular rosacea lesions, but different from that of erythematotelangiectatic and phymatous rosacea lesions (Fig 7A–7C).

### 3.7. Infiltration of Th1/Th17 cells

Th1/Th17 pathways regulate several critical pathways implicated in acne [10, 23] and rosacea [9] pathogenesis. However, the analysis of 22 kinds of immune cells through CIBERSORT

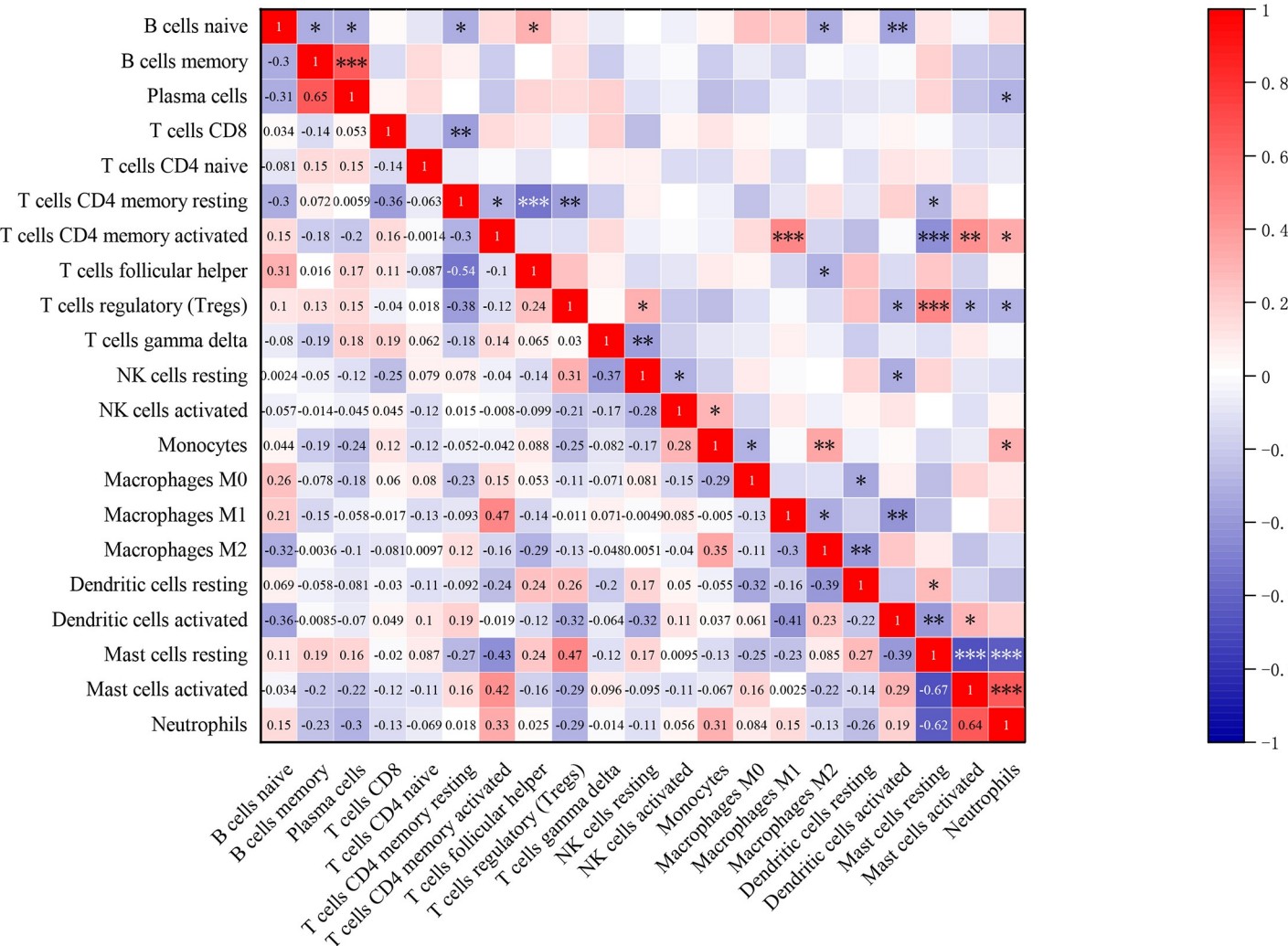

**Fig 4. The correlation in infiltration among 21 subpopulations of immune cells (except eosinophils).** The darker the color, the stronger the correlation. Blue graph represents negative correlation between two cell types, whereas red graph represents positive correlation between two cell types. The correlation coefficients (r) are shown in the squares. *$P<0.05$, **$P<0.01$, ***$P<0.001$.

algorithm did not include Th1 and Th17 cells. Against this background, we analyzed the expression of genes associated with Th1-related molecules (IL2, IL12RB1), Th17-related cytokines (IL17A, IL17F) and the infiltration of Th1/Th17 cells. We found that compared to non-lesional sites of acne patients, there was an up-regulated expression of all the Th1/17-related cytokines in the acne lesions. In addition, IL12RB1 was over-expressed in the acne lesions, relative to rosacea lesions (Fig 8). Interestingly, IL12RB1, IL17A and IL17F were under-expressed less in non-lesional skins of acne patients, relative to healthy skins.

## 4. Discussion

Acne and rosacea are common chronic inflammatory skin diseases with several pathological distinctions. Clinically, acne presents with high seborrhoea, open and closed comedones (inflammatory lesions) and papules, pustules, nodules and cysts (inflammatory lesions). In addition, close to 20% of patients develop severe acne, which is accompanied by scaring of the skin [6]. The acne lesions occur in face, neck, upper chest, shoulders and back, generally in

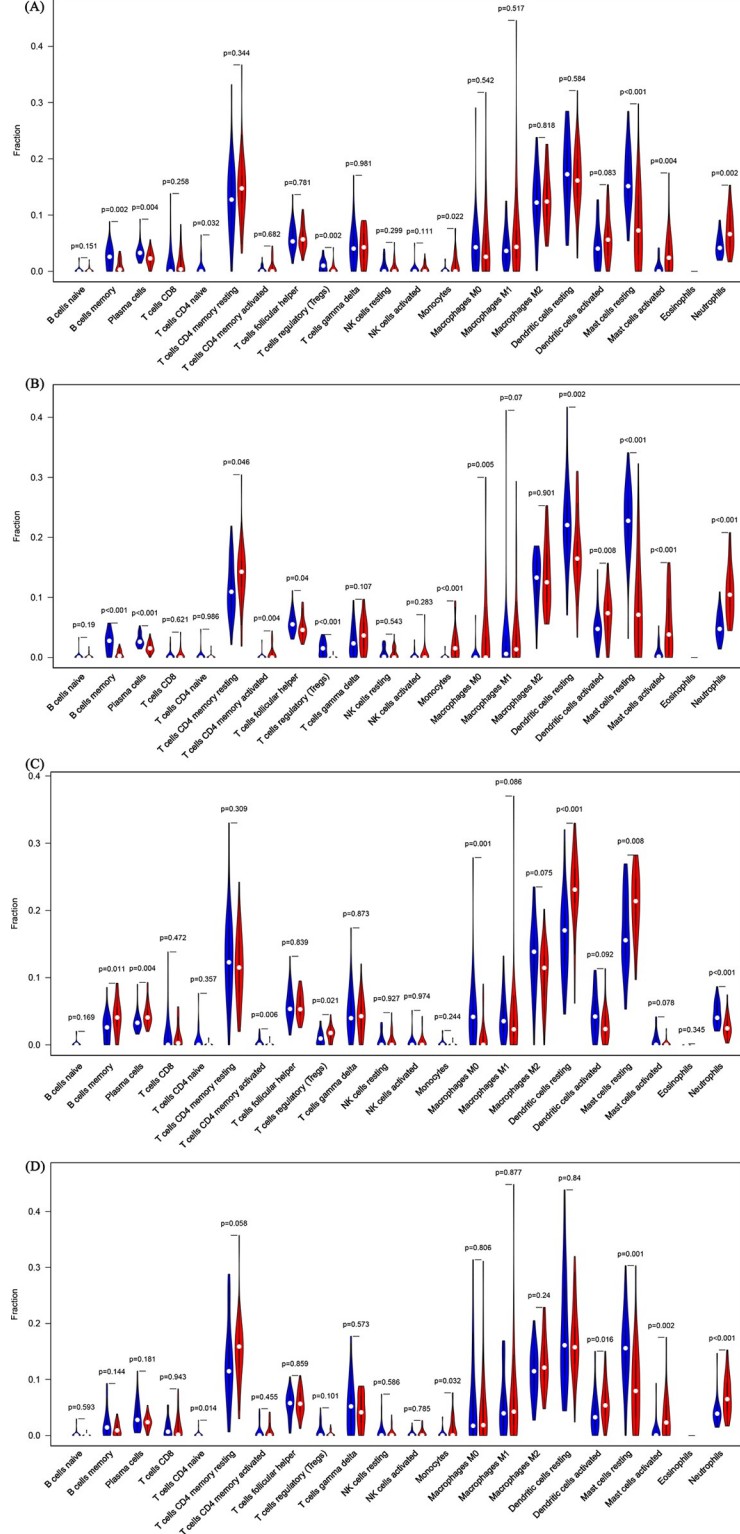

**Fig 5.** The infiltration pattern of immune cells between (A) Acne lesions (red) vs. healthy individuals (blue), (B) Acne lesions (red) vs. non-lesional skin sites of acne patients (blue), (C) Non-lesional skin of acne patients (red) vs. healthy individuals (blue), (D) Acne lesions (red) vs. rosacea lesions (blue). (*P* < 0.05).

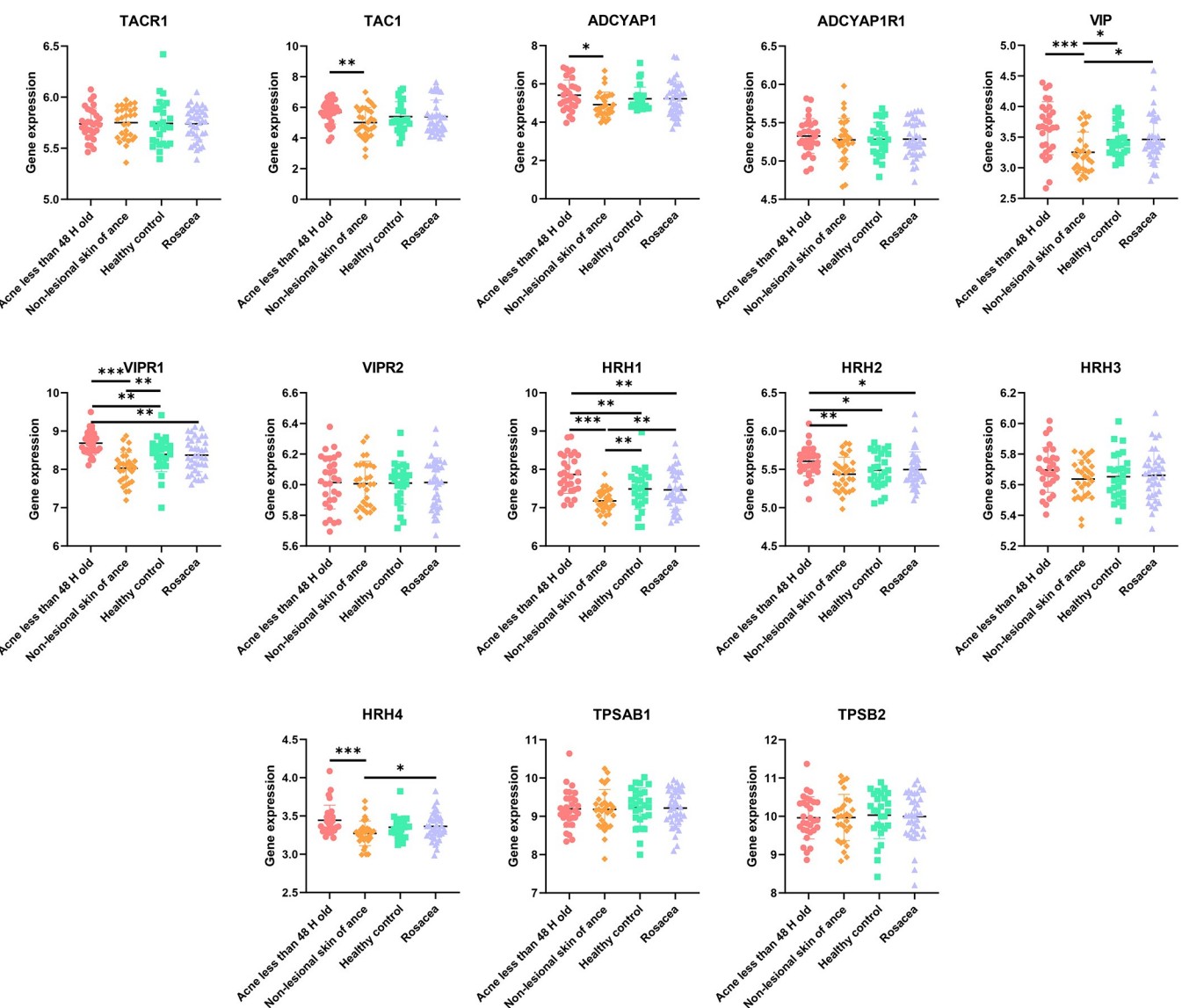

**Fig 6. The correlation between infiltration of mast cells and secretion of other inflammatory molecules among the study 4 groups.** (Adjust *P*< 0.004 was considered statistically significant. *Adjust *P*<0.004, **Adjust *P*<0.001, *** Adjust *P*<0.0001).

areas with high pilosebaceous units [24]. On the other hand, rosacea syndrome has four distinct clinical subtypes; erythematotelangiectatic, papulopustular, phymatous and ocular rosacea [25]. Erythematotelangiectatic rosacea, the most common rosacea sub-type, is characterized by flushing and persistent central facial erythema with or without telangiectasia. Papulopustular rosacea presents with persistent central facial erythema with transient central facial papules or pustules or both. Phymatous rosacea is characterized by thickening of the skin and large, irregular and surface nodularities. Patients with ocular rosacea have the sensation of foreign body in the eye, burning or stinging, dryness, itching, ocular photosensitivity, blurred vision, telangiectasia of the sclera or other parts of the eye, or periorbital edema [25]. Treatment for erythematotelangiectasia mainly includes topical (brimonidine, oxymetazoline) and light/laser therapies (intense pulsed light, potassium titanyl phosphate and pulsed dye laser). On the other hand, topical treatment with ivermectin, azelaic acid, metronidazole,

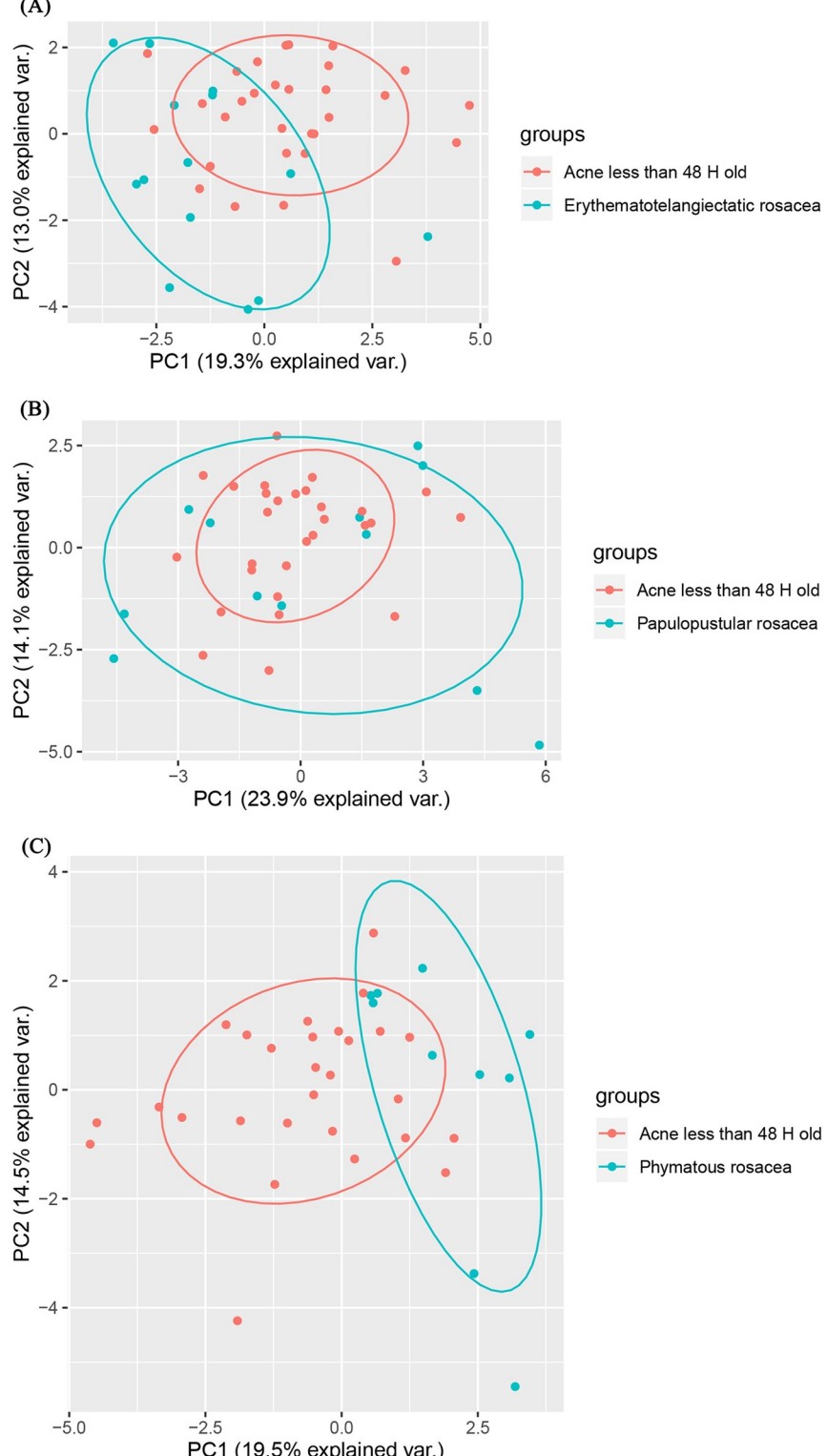

**Fig 7. The comparative infiltration of immune cells between acne lesions and rosacea subtypes.**

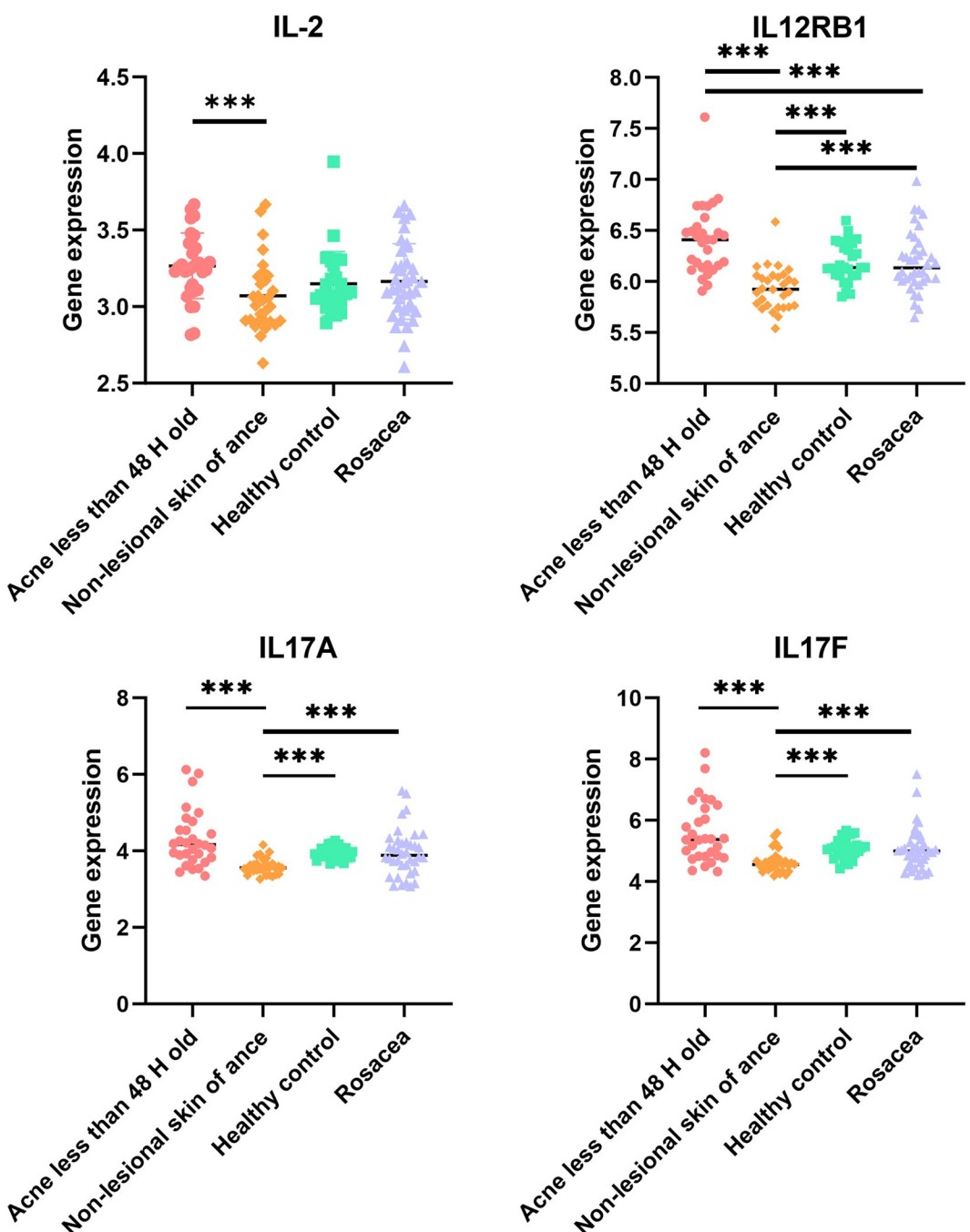

**Fig 8. Expression of genes for Th1/Th17-related molecules.** (Adjust $P< 0.004$ was considered statistically significant. *Adjust $P<0.004$, **Adjust $P<0.001$, ***Adjust $P<0.0001$).

sodium sulfacetamide, clindamycin and brimonidine, or oral therapy with doxycycline, mino-cycline, azithromycin, bactrim and sotretinoin are recommended for papulopustular rosacea. Phymatous rosacea is often treated with topical retinoids, oral medicine (similar with oral therapies for papulopustular) and procedural therapies of $CO_2$ and erbium lasers, electrosurgical and radiofrequency rays. For ocular rosacea, topical, oral and light therapies have been used [26]. The management of acne includes topical (i.e., antibiotics, retinoids, azelaic acid, salicylic

acid, benzoyl peroxide, dapsone, antiandrogens, ASC-J9 cream, topical anticholinergic agent), systemic (i.e., retinoids, antibiotics, hormonal, insulin-like growth factor-1 inhibitors, 5-lipoxygenase inhibitor, acetyl-coenzyme A carboxylase inhibitors, phosphodiesterase inhibitors, IL1β inhibitors, vitamin D analogs, dapsone) and light/laser (i.e., red, blue and broadband light; pulsed dye lasers and infrared rays; photodynamic therapy with methylaminolevulinic and aminolevulinic acids) treatments [27]. Besides the many differences between acne and rosacea, both diseases arise from inflammatory processes. This explains why historically, the two diseases were classified in the same category [28]. Acne and rosacea mainly occur on the face, and present with similar clinical manifestations such as erythema, papules, pustules and nodules on the skin. Pathogenetically, both acne and rosacea have been associated with *Staphylococcus epidermis* infection. Topical (clindamycin, benzoyl peroxide, azelaic acid, retinoids cream) and oral (minocycline, retinoids, etc.) therapies have been used for the treatment of the two diseases. Therefore, we characterized immune cells infiltration profile of acne and rosacea to deepen our understanding on the precise pathogenesis of the two diseases.

Even though the etiologies of acne and rosacea are not well understood, innate and adaptive immunity (inflammation) have been implicated in the pathogenesis of the two diseases. Acne and rosacea share numerous inflammatory processes. For instance, activation of microbes mediated Toll-like receptor (TLR) 2 and NOD-like receptor 3 (NLRP3) inflammasome and subsequent IL1β and TNF-mediated inflammation has been reported in both diseases [22, 29, 30]. In addition, microbe driven polarization of Th1/Th17 [9, 31], infiltration of macrophages and mast cells [22, 32, 33], has also been observed in both conditions. However, this study uncovered several differences in the infiltration pattern of immune cells between acne and rosacea. In particular, compared with rosacea patients, there was a higher infiltration of neutrophils, monocytes and activated mast cells and over-expression of Th1/Th17-related molecules in acne lesions. Contrarily, we observed lower infiltration of naive CD4+ T cells, plasma cells, memory B cells and resting mast cells in acne lesions, relative to rosacea lesions. The infiltration of mast cells in areas adjacent to the sebaceous glands of acne lesion has also been previously reported [34]. Intriguingly, the over-expression of molecules associated with activation and degranulation of mast cells further supported the infiltration of activated mast cells in acne lesions. Meanwhile, neutrophils infiltrated in the acne papules and papulopustular rosacea lesions but not erythematotelangiectatic and phymatous rosacea lesions. Research shows that infiltration of plasma cells in later stages of acne development was associated with scarring. Plasma cells have been observed in papulopustular and phymatous rosacea lesions [8, 9], consistent with our findings. PCA cluster analyses revealed that the infiltration pattern of immune cells in the acne lesions was comparable to that of papulopustular lesions. Although Th1/Th17 pathway participates in the pathogenesis of acne and rosacea, we found the effect is greater in acne than rosacea.

Skin is the first line of defense against entry of pathogenic organisms. Any physical, chemical and microbial damage of the skin results in the recruitment of several types of immune cells to the damage site. The resultant inflammation restores the homeostasis of the region [35]. Under normal conditions, immune cells only account for 7% of the cells in skin, which constitutes of 3.78% Langerhans cells (a subset of resident macrophages in skin), 0.45% NK cells, 0.24% monocytes, 0.79% dendritic cells, 0.41% T cells and 1.33% other immune cells such as innate lymphoid cells (ILC), neutrophils and macrophages [36]. Notably, 80% of the skin resident T cells are memory T cells [37]. Meanwhile, CD4+ and CD8+ memory T cells account for 83% and 17% respectively, of the T cells [38], whereas Tregs account for 5–10% of the skin resident T cells.

We further revealed the high infiltration of neutrophils, monocytes, activated mast cells but low infiltration of Tregs in acne lesions, manifesting a state of immunity enhancement relative

to skin of healthy individuals. In addition, we observed lower infiltration of activated memory CD4+ T cells, plasma cells, memory B cells, M0 macrophages, neutrophils, resting mast cells but higher infiltration of Tregs and resting dendritic cells in non-lesional skin of acne patients presenting a certain degree of immunosuppression relative to skin of healthy individuals. High infiltration of CD3+ and CD4+ T cells and macrophages as well as over-expression of IL1, vascular adhesion molecule 1 and E-selectin have been reported in skin in the periphery of acne lesions [39]. This implies that even though immune events occur in both non-lesional areas of acne patients, the response is significantly low when compared with acne lesions. This was demonstrated by the low levels of infiltrated activated memory CD4+ T cells, plasma cells, memory B cells, M0 macrophages, neutrophils and resting mast cells, under-expression of Th1/17 related molecules such as IL2, IL12RB1, IL17A, IL17F but over-infiltration of Tregs and resting dendritic cells in non-lesional skin of acne patients. This suggests of immunosuppression or immune insensitivity in the non-lesional skin areas of acne patients, which may explain the non-occurrence of papules in such areas. In addition, the immune phenomenon may result from increased movement of immune cells from tissues adjacent to the acne to the lesioned areas. Wen Yue Jiang et al found the migration of mast cells from normal skin regions to psoriasis lesions [40]. In a separate study, Scott [41] hypthesised that when localized infection or inflammation occurs in the skin, resident memory CD4+ T cells in the adjacent areas migrate to the inflammatory sites. Patrick et al. [42] also reported that B cells and neutrophils can selectively migrate through the fibroblast barrier to inflammatory sites. These findings theoretically support our findings stated in the proceeding sections.

Inflammation and autoimmunity are associated with increased infiltration of Th17 cells, but decrease in migration Treg cells to the affected site [43, 44]. A higher Th17/Treg ratio has been observed in hidradenitis suppurativa (HS)/acne inversa, comparable with our findings. This study also revealed strong positive correlations between infiltration of activated memory CD4+ T cells and M1 macrophages, neutrophils and activated mast cells, activated memory CD4+ T cells and activated mast cells, but a negative correlation between resting memory CD4 + T cells and follicular helper T cells. Thus, the complex interactions between immune cells influence the pathogenesis of acne and rosacea.

## 5. Conclusions

Both acne and rosacea are common chronic inflammatory skin diseases, whose pathogeneses are associated with infiltration of immune cells. Overall, acne lesions exhibit higher infiltration of neutrophils, monocytes, activated mast cells but lower infiltration of naive CD4+ T cells, plasma cells, memory B cells, resting mast cells, relative to normal skin adjacent to the acne lesions, that of healthy individuals and of rosacea lesions. Lower infiltration of Tregs was also observed in acne lesions, relative to normal skin of acne patients and healthy individuals. For normal skin areas of acne patients, there is lower infiltration of activated memory CD4+ T cells, plasma cells, memory B cells, M0 macrophages, neutrophils and resting mast cells but higher infiltration of Tregs and resting dendritic cells, relative to skins of healthy individuals. Among the 3 rosacea disease subtypes, the immune infiltration pattern in papulopustular rosacea is comparable to that of acne. Besides, Th1 and Th17 cells are over-expressed in acne lesions, relative to normal skins of acne patients.

## Supporting information

**S1 File.**
(DOCX)

## Acknowledgments

Lu Yang and Yan-hong Shou have equal contributions to this paper and should be treated as co-first authors. The authors thank DR. Zhi-wen Luo and DR. Yu Han for technical consulting in data processing.

## Author Contributions

**Conceptualization:** Lu Yang, Yan-Hong Shou, Yong-Sheng Yang.

**Data curation:** Lu Yang, Yong-Sheng Yang, Jin-Hua Xu.

**Formal analysis:** Lu Yang.

**Investigation:** Lu Yang, Yan-Hong Shou.

**Methodology:** Lu Yang.

**Software:** Lu Yang, Yan-Hong Shou.

**Supervision:** Yong-Sheng Yang, Jin-Hua Xu.

**Validation:** Yong-Sheng Yang, Jin-Hua Xu.

**Visualization:** Lu Yang.

**Writing – original draft:** Lu Yang.

**Writing – review & editing:** Yong-Sheng Yang, Jin-Hua Xu.

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
