## [Decision Letter · Decision Letter 0]

23 Dec 2020

PONE-D-20-30237

Elucidating the immune infiltration in acne and its comparison with rosacea by integrated bioinformatics analysis

PLOS ONE

Dear Dr. Xu,

Thank you for submitting your manuscript to PLOS ONE. After careful consideration, we feel that it has merit but does not fully meet PLOS ONE’s publication criteria as it currently stands. Therefore, we invite you to submit a revised version of the manuscript that addresses the points raised during the review process.

As suggested by the reviewer,  this studies exclusively focus on immune cells, the potential involvement of other types of cells, such as epithelial cells, endothelial cells etc, were not considered. Please provide a better discussion about their potential involvement and carefully correct the typos and grammatical errors.

We look forward to receiving your revised manuscript.

Kind regards,

Deyu Fang, Ph.D.

Academic Editor

PLOS ONE

Journal Requirements:

2.) Thank you for stating the following financial disclosure:

'The funders had no role in study design, data collection and analysis, decision to publish, or preparation of the manuscript.'

3.) PLOS requires an ORCID iD for the corresponding author in Editorial Manager on papers submitted after December 6th, 2016. Please ensure that you have an ORCID iD and that it is validated in Editorial Manager. To do this, go to ‘Update my Information’ (in the upper left-hand corner of the main menu), and click on the Fetch/Validate link next to the ORCID field. This will take you to the ORCID site and allow you to create a new iD or authenticate a pre-existing iD in Editorial Manager. Please see the following video for instructions on linking an ORCID iD to your Editorial Manager account: https://www.youtube.com/watch?v=_xcclfuvtxQ

Reviewers' comments:

Reviewer's Responses to Questions

**Comments to the Author**

1. Is the manuscript technically sound, and do the data support the conclusions?

Reviewer #1: Partly

2. Has the statistical analysis been performed appropriately and rigorously? 

Reviewer #1: Yes

3. Have the authors made all data underlying the findings in their manuscript fully available?

Reviewer #1: Yes

4. Is the manuscript presented in an intelligible fashion and written in standard English?

Reviewer #1: Yes

5. Review Comments to the Author

Reviewer #1: ALSO PROVIDED AS ATTACHMENT

Review for PLOS One Manuscript

“Elucidating the immune infiltration in acne and its comparison with rosacea by integrated bioinformatics analysis”

Submitted 12/22/20

The manuscript by Yang and colleagues investigates differential immune infiltration in common inflammatory conditions of facial skin. Using bioinformatics approaches, the authors seek out how these different pathologies may recruit, promote, or repress different immune cell subsets. Important questions the study attempts to answer are (1) How does immune cell infiltration differ between common conditions of acne vulgaris and rosacea, (2) how does immune cell infiltration differ in the lesions of acne patients compared to the unaffected skin in those same patients, and (3) are there immune components highly specific to acne lesions that may serve as immunotherapeutic targets. The authors draw upon publicly available microarray datasets of whole skin biopsies from patients with acne vulgaris, different forms of rosacia, and healthy controls. The authors first remove batch effects and normalize between datasets then apply the commonly used CIBERSORT algorithm in order to deconvolute the bulk microarray data in order to gain insight into different immune cell infiltrates. Key findings unique to active acne lesions include higher neutrophil, monocyte, and activated mast cell infiltration relative to other conditions. Given the previous characterization of Th1/Th17 imbalance in inflammatory skin conditions this study finds that the relative recruitment of T cells is largely unaltered. Interestingly, the authors also find that non-lesional skin of acne patients typically have a higher rate of Treg infiltration. Because the CIBERSORT algorithm does not parse out different T cell subsets, the authors conclude by assessing relative expression of T helper cell subsets, Th1 and Th17 related molecules, and find that they are elevated, agreeing with prior findings. The study is fundamentally strong and seeks to answer important questions in the field. With some additional analysis and clarification of comments below, I would recommend the study for publication.

Major Comments

1. The authors use the CIBERSORT algorithm to great effect and it reveals much about the differences in immune cell recruitment between different inflammatory pathologies about the skin. They fail, however, to go further with the dataset they have at their disposal. For example, the study identifies activated mast cells as one of the populations recruited higher in lesional acne relative to other conditions and then goes on in the discussion (lines 258-262) and mentions molecules implicated in the recruitment of mast cells. The authors could simply screen expression of these molecules in the analyzed datasets to determine whether they are partially responsible for immune cell recruitment.

2. Another fundamental problem with the study is that it largely focuses on immune cell infiltration of these different pathologies, but then only reviews the relative proportions of immune cells to each other. The study needs to also cover what proportion of the samples were immune cells relative to other cell types (eg epithelial cells, endothelial cells etc.). This can be accomplished by either contacting originators of the original datasets for the most accurate data, or if not possible they can contact other clinicians working in the field or others with relevant animal models.

Minor Comments

1. Lines 83-84 are these lesions and non-lesions from the same patients?

2. Figure 2A and 2B include 2-dimensional PCA plots for ease of visualization

3. Figure 2A and 2B include %variance explained by each principal component

4. Lines 108-109 p-value attained from differential expression or from CIBERSORT algorithm? Please specify

5. Line 122 typo is “valuated” should be “evaluated”

6. Does CIBERSORT tell you what proportion of TOTAL cells biopsied are immune or hematopoietic? Here we see relative proportions of immune subsets which may be misleading if there are vastly different proportions of immune infiltration to begin with.

7. Figures 3A and 3B need labelling to identify which datasets correspond to which groups of patients, similar to figures 3a and 3b

8. 3C and 3D should flip cluster heatmaps so that “healthy patients” are in the same side of the figure, this can be done easily in R by ordering the samples how they naturally cluster, then turning off the cluster columns in the pheatmap function

9. Some more intro as to why it is important to compare acne and rosacea… are they commonly associated/treated?

10. Figure 4 needs to depict what the numeric values are in reference to. To me it seems as though they are R^2 values but it should be clearer. Additionally, p-values for the correlations referred to in lines 166-168. Lastly, is this correlation matrix constructed from within the acne <48hrs group? The text speaks about a comparison between acne <48hrs and healthy controls which makes it confusing as to what data is used to construct the correlation matrix.

11. What data is the PCA cluster analysis based off? If it is based off total gene expression data and not just the CIBERSORT data, then it is misleading to stat that it is comparison in immune cell infiltration.

12. Lines 199-200 comment on infiltration of Th1/Th17 cells but the authors are looking at expression of molecules that are expressed by multiple cell-types therefore it is inaccurate to make the claim that they are assessing Th1/Th17 infiltration

13. From 271-285 it is clear that skin immunology and facial skin immunology at the steady state are not discussed. This can be covered either in the introduction or the discussion but the authors should discuss what sorts of immune cells are normally present in the skin or in the facial skin in particular and in what proportions. This is covered partially through their computational work but they should also cover the prior literature.

14. Another possibility to explain the dearth of activated immune cells in non-lesional skin of acne patients is the possibility that immune cells that are normally evenly distributed throughout the skin (eg in healthy patients examined in the study) traffic into lesions and leave the non-lesional skin immune depleted. The authors can cover this in their discussion if appropriate.

6. PLOS authors have the option to publish the peer review history of their article (what does this mean?). If published, this will include your full peer review and any attached files.

Reviewer #1: No

---

## [Author Response · Author response to Decision Letter 0]

25 Feb 2021

Dear reviewer:

Thank you very much for your valuable suggestions for us. We have carefully read all issues mentioned in your comments, and revised our article after conscientious consideration. Here, we will reply to your comments point by point. We look forward to discussing with you, and always look forward to your advice！

Major Comments

1.The authors use the CIBERSORT algorithm to great effect and it reveals much about the differences in immune cell recruitment between different inflammatory pathologies about the skin. They fail, however, to go further with the dataset they have at their disposal. For example, the study identifies activated mast cells as one of the populations recruited higher in lesional acne relative to other conditions and then goes on in the discussion (lines 258-262) and mentions molecules implicated in the recruitment of mast cells. The authors could simply screen expression of these molecules in the analyzed datasets to determine whether they are partially responsible for immune cell recruitment.

Reply: Thank you for your advice! Our study identified activated mast cells as one of the populations recruited higher in acne lesions relative to other conditions. In the recruitment of mast cells, a lot of molecules are involved such as substance P (SP), pituitary adenylate cyclase-activating polypeptide(PACAP), vasoactive intestinal peptide(VIP), histamine and tryptase. We simply screened the expressions of the genes of these molecules in the analyzed datasets to determine whether they are partially responsible for mast cell recruitment. Among them, TAC1 and TACR1 encode SP and its receptor respectively. ADCYAP1 and ADCYAP1R1 encode PACAP and its receptor respectively. VIP and its receptor are encoded by VIP and VIPR1, VIPR2. Histamine is encoded by HRH1-4. TPSAB1 and TPSB2 encode tryptase. As shown in Fig 6, there was an upregulated expression of SP and PACAP in acne lesions relative to normal skins from the same acne patients. VIP and histamine were expressed highest in acne lesions, moderate in rosacea lesions and skin of healthy patients and least in normal skins of the acne patients. There were no significant differences in the expressions of tryptase genes among the four groups. The expressions of these molecules involved in mast cells’ activation and degranulation evaluated by us supported that the infiltration of activated mast cells is increased in acne lesions.

2.Another fundamental problem with the study is that it largely focuses on immune cell infiltration of these different pathologies, but then only reviews the relative proportions of immune cells to each other. The study needs to also cover what proportion of the samples were immune cells relative to other cell types (eg epithelial cells, endothelial cells etc.). This can be accomplished by either contacting originators of the original datasets for the most accurate data, or if not possible they can contact other clinicians working in the field or others with relevant animal models.

Reply: Thank you very much for your valuable suggestions. In our article, the description of CIBERSORT algorithm was not clear enough. In addition, we did not interpret the results of the comparison of immune cell infiltration between samples in detail, which may make readers misunderstand that we only compared the relative proportions of immune cells to each other. Therefore, in our modification process, we added the description of the algorithm we used and our results as well as figures. We used CIBERSORT algorithm to deconvolute all the normalized gene expression profiles, deconvolution algorithms means that the expression level of a gene in a sample is a linear combination of the expression level of the gene in different cell subpopulations and the weight of cell fraction. Therefore, the “absolute CIBERSORT score” we got from CIBERSORT algorithm represented the cell fraction weight of each immune cell in different samples, which can reflect the absolute content of the immune cells in each sample. In our study, CIBERSORT was run in the mode of absolute quantification (method: sig score), rather than the relative mode used in the original publication. That means absolute CIBERSORT scores were used to reflect the absolute content of 22 kinds of immune cells in each sample, different scores between different samples can be directly used for comparison, just like normalized gene expression profiles can be directly compared between different samples. According to the absolute CIBERSORT scores, we calculated when the total infiltration was 100%, the relative proportions of each kind of infiltrated cells, in order to show the proportion of immune cells in each sample more intuitively (Fig 3 A-B), all of the other analyses of immune infiltration compared absolute CIBERSORT scores among samples. 

Thank you for your suggestion that the study needs to also cover what proportion of the samples were immune cells relative to other cell types (eg epithelial cells, endothelial cells etc.). At present, there are many methods that can simultaneously analyze immune cells and non-immune cells (including epithelial cells, endothelial cells, fibroblasts, melanocytes, etc.) and compare their relative proportions such as Xcell method based on ssGSEA, however, the classification of immune cells in Xcell is not detailed enough. For example, mast cells, dendritic cells, memory CD4+ T cells and NK cells in activated and resting state cannot be analyzed separately, in addition, it can’t analyze the immune infiltration of follicular helper T cells and gamma delta T cells, but can only describe the proportions of immune cells relative to non-immune cells, which couldn’t help us to clarify the immune infiltration in acne and its comparison with rosacea in more detail. If the proportions of immune cells relative to non-immune cells are used as a reference for the absolute contents of immune cells in the sample, we think the absolute CIBERSORT score of immune cells obtained by CIBERSORT algorithm can reflect them more accurately. In addition, although the proportions of non-immune cells in the samples were often analyzed in tumor, such as epithelial cells were enriched in many tumors, keratinocytes were enriched in squamous cell carcinoma, mesangial cells were enriched in renal cell carcinoma, chondrocytes were enriched in sarcomas, etc., which could help to determine the origin and nature of the tumor. However, in the pathogenesis of acne, the role of non-immune cells such as endothelial cells, endothelial cells, fibroblasts and melanocytes is not clear. To sum up, the significance of comparing what proportion of the samples were immune cells relative to other cell types is limited. If you still suggest that we should cover the proportion immune cells relative to other cell types, we are looking forward to communicating and discussing with you to determine what kinds of immune cells and non-immune cells we should cover. Please allow us to express our appreciation to you again.

Minor Comments

Lines 83-84 are these lesions and non-lesions from the same patients?

Reply: Yes, these lesions and non-lesions are from the same patients, in this revision, we added an explanation about it.

Figure 2A and 2B include 2-dimensional PCA plots for ease of visualization

Reply: Thanks for your suggestion! We replaced the three-dimensional PCA plot with the two-dimensional PCA plot through “ggplot2” package in R software.

Figure 2A and 2B include %variance explained by each principal component

Reply: We have made change that in the new Figure 2A and 2B, %variance explained by each principal component was added. Thank you!

Lines 108-109 p-value attained from differential expression or from CIBERSORT algorithm? Please specify

Reply: We chose the samples with P-value < 0.05 from CIBERSORT algorithm, and obtained the 22 kinds of immune cells’ composition. In our revision, we made an explanation here. Thank you!

Line 122 typo is “valuated” should be “evaluated”

Reply: We have changed the writing. Thank you!

Does CIBERSORT tell you what proportion of TOTAL cells biopsied are immune or hematopoietic? Here we see relative proportions of immune subsets which may be misleading if there are vastly different proportions of immune infiltration to begin with.

Reply: CIBERSORT didn’t tell us what proportion of TOTAL cells biopsied are immune or hematopoietic, this question may only be answered by single cell sequencing. However, the absolute CIBERSORT score we compared among samples did not reflect the relative proportions of immune subsets, but the cell fraction weight of each kind of immune cell in each sample. It was similar to the meaning of "the proportions of immune infection to begin with". Therefore, these results will not be affected by different total cells biopsied among samples.

Figures 3A and 3B need labelling to identify which datasets correspond to which groups of patients, similar to figures 3a and 3b

Reply: Thanks! Following your advice, we added labels to Figures 3 A and 3 B.

3C and 3D should flip cluster heatmaps so that “healthy patients” are in the same side of the figure, this can be done easily in R by ordering the samples how they naturally cluster, then turning off the cluster columns in the pheatmap function

Reply: Following your advice, we adjusted the order in Figures 3 C and 3 D, now “healthy patients” are in the same side of the figure.

Some more intro as to why it is important to compare acne and rosacea… are they commonly associated/treated?

Reply: Although there are many differences between the two diseases, acne and rosacea are common inflammatory processes historically classified in the same disease category (https://pubmed.ncbi.nlm.nih.gov/27416314/). Acne and rosacea mainly occur on the face, and present with similar clinical manifestations such as erythema, papules, pustules and nodules on the skin. Pathogenetically, both acne and rosacea have been associated with Staphylococcus epidermis infection. Topical (clindamycin, benzoyl peroxide, azelaic acid, retinoids cream) and oral (minocycline, retinoids, etc.) therapies have been used for the treatment of the two diseases. Therefore, we characterized immune cells infiltration profile of acne and rosacea to deepen our understanding on the precise pathogenesis of the two diseases.

Figure 4 needs to depict what the numeric values are in reference to. To me it seems as though they are R^2 values but it should be clearer. Additionally, p-values for the correlations referred to in lines 166-168. Lastly, is this correlation matrix constructed from within the acne <48hrs group? The text speaks about a comparison between acne <48hrs and healthy controls which makes it confusing as to what data is used to construct the correlation matrix.

Reply: The numeric values represented the correlation coefficient r values, and we have redrawn Figure 4 by “OriginPro” software containing both correlation coefficients and P-values. This correlation matrix constructed from within the acne <48hrs group. In this revision, we explained them in the article. 

What data is the PCA cluster analysis based off? If it is based off total gene expression data and not just the CIBERSORT data, then it is misleading to stat that it is comparison in immune cell infiltration.

Reply: Thank you very much for your advice. PCA cluster analysis results were based off the CIBERSORT data, but we didn't have a clear description here. In the revision, we added an explanation.

Lines 199-200 comment on infiltration of Th1/Th17 cells but the authors are looking at expression of molecules that are expressed by multiple cell-types therefore it is inaccurate to make the claim that they are assessing Th1/Th17 infiltration

Reply: When we evaluated the immune infiltration of Th1/Th17 cells, we focused on many molecules that were expressed by multiple cell-types, this was unreasonable. In the process of modification, we only compared the molecules which were specificly correlated with Th1/17 cells, including IL2, IL12RB1, IL17A and IL17F. We think it can partly explain the infiltration of Th1/Th17 cells in acne and rosacea. Although it is known that Th1/Th17 pathway plays a vital role in the pathogenesis of acne and rosacea, our results showed higher Th1/Th17-related molecules expression level in acne compared with rosacea. Interestingly, IL12RB1, IL17A, IL17F expressed less in non-lesional skin in acne patients compared with healthy individuals.

From 271-285 it is clear that skin immunology and facial skin immunology at the steady state are not discussed. This can be covered either in the introduction or the discussion but the authors should discuss what sorts of immune cells are normally present in the skin or in the facial skin in particular and in what proportions. This is covered partially through their computational work but they should also cover the prior literature.

Reply: Thank you very much, you suggested that we should discuss skin immunology and facial skin immunology at the steady state in the prior literature. However, we have not found any articles specifically describing the composition of immune cells in normal facial skin. Therefore, we only discussed the composition of immune cells in normal skin, and we added the following to the discussion section. 

Skin is the first line of defense against entry of pathogenic organisms. Any physical, chemical and microbial damage of the skin results in the recruitment of several types of immune cells to the damage site. The resultant inflammation restores the homeostasis of the region ( https://pubmed.ncbi.nlm.nih.gov/30429578/). Under normal conditions, immune cells only account for 7% of the cells in skin, which constitutes of 3.78% Langerhans cells (a subset of resident macrophages in skin), 0.45% NK cells, 0.24% monocytes, 0.79% dendritic cells, 0.41% T cells and 1.33% other immune cells such as innate lymphoid cells (ILC), neutrophils and macrophages (https://pubmed.ncbi.nlm.nih.gov/31046232/). Notably, 80% of the skin resident T cells are memory T cells (https://pubmed.ncbi.nlm.nih.gov/16547281/). Meanwhile, CD4+ and CD8+ memory T cells account for 83% and 17% respectively, of the T cells (https://pubmed.ncbi.nlm.nih.gov/25787765/), whereas Tregs account for 5-10% of the skin resident T cells.

Another possibility to explain the dearth of activated immune cells in non-lesional skin of acne patients is the possibility that immune cells that are normally evenly distributed throughout the skin (eg in healthy patients examined in the study) traffic into lesions and leave the non-lesional skin immune depleted. The authors can cover this in their discussion if appropriate.

Reply: Thank you very much for making us aware of this possibility, it’s interesting and reasonable! Following your advice, this opinion was supplemented to the discussion section. Although we didn’t find any evidence of immune cell migration from non-lesional skin to lesional skin in acne patients in the prior literature, we found some evidences of immune cell migration in other cases. Our description was as follows.

In addition, the immune phenomenon may result from increased movement of immune cells from tissues adjacent to the acne to the lesioned areas. Wen Yue Jiang et al found the migration of mast cells from normal skin regions to psoriasis lesions (https://pubmed.ncbi.nlm.nih.gov/11737436/). In a separate study, Scott (https://pubmed.ncbi.nlm.nih.gov/23215646/) hypthesised that when localized infection or inflammation occurs in the skin, resident memory CD4+ T cells in the adjacent areas migrate to the inflammatory sites. Patrick et al. (https://pubmed.ncbi.nlm.nih.gov/19394331/) also reported that B cells and neutrophils can selectively migrate through the fibroblast barrier to inflammatory sites. These findings theoretically support our findings stated in the proceeding sections.

---

## [Editor Report · Decision Letter 1]

3 Mar 2021

Elucidating the immune infiltration in acne and its comparison with rosacea by integrated bioinformatics analysis

PONE-D-20-30237R1

Dear Dr. Xu,

We’re pleased to inform you that your manuscript has been judged scientifically suitable for publication and will be formally accepted for publication once it meets all outstanding technical requirements.

Kind regards,

Deyu Fang, Ph.D.

Academic Editor

PLOS ONE
---

## [Editor Report · Acceptance letter]

16 Mar 2021

PONE-D-20-30237R1 

Elucidating the immune infiltration in acne and its comparison with rosacea by integrated bioinformatics analysis 

Dear Dr. Xu:

I'm pleased to inform you that your manuscript has been deemed suitable for publication in PLOS ONE. Congratulations! Your manuscript is now with our production department. 

Kind regards, 

on behalf of

Dr. Deyu Fang 

Academic Editor

PLOS ONE